# Contrastive Context-Speech Pretraining for Expressive Text-to-Speech Synthesis

Yujia Xiao
The Chinese University of Hong Kong
Hong Kong, China
yujiaxiao@link.cuhk.edu.hk

Xi Wang
Microsoft
Beijing, China
xwang@microsoft.com

Xu Tan
Microsoft
Beijing, China
xuta@microsoft.com

Lei He
Microsoft
Beijing, China
helei@microsoft.com

Xinfa Zhu
Microsoft
Beijing, China
xfzhu@mail.nwpu.edu.cn

Sheng Zhao
Microsoft
Beijing, China
szhao@microsoft.com

Tan Lee*
The Chinese University of Hong Kong
Hong Kong, China
tanlee@ee.cuhk.edu.hk

## Abstract

The latest Text-to-Speech (TTS) systems can produce speech with voice quality and naturalness comparable to human speech. Yet the demand for large amount of high-quality data from target speakers remains a significant challenge. Particularly for long-form expressive reading, target speaker's training speech that covers rich contextual information are needed. In this paper a novel design of context-aware speech pre-trained model is developed for expressive TTS based on contrastive learning. The model can be trained with abundant speech data without explicitly labelled speaker identities. It captures the intricate relationship between the speech expression of a spoken sentence and the contextual text information. By incorporating cross-modal text and speech features into the TTS model, it enables the generation of coherent and expressive speech, which is especially beneficial when there is a scarcity of target speaker data. The pre-trained model is evaluated first in the task of Context-Speech retrieval and then as the integral part of a zero-shot TTS system. Experimental results demonstrate that the pretraining framework effectively learns Context-Speech representations and significantly enhances the expressiveness of synthesized speech. Audio demos are available at: https://ccsp2024.github.io/demo/.

## CCS Concepts

• **Human-centered computing** → **Human computer interaction (HCI)**; **Human computer interaction (HCI)**; • **Computing methodologies** → **Artificial intelligence**; **Neural networks**; **Artificial intelligence**.

## Keywords

Contrastive Learning, Pretraining Model, Speech Synthesis, Contextual Modeling, Cross-modal Learning

---

*Tan Lee is the Corresponding Author.

**ACM Reference Format:**
Yujia Xiao, Xi Wang, Xu Tan, Lei He, Xinfa Zhu, Sheng Zhao, and Tan Lee. 2024. Contrastive Context-Speech Pretraining for Expressive Text-to-Speech Synthesis. In *Proceedings of the 32nd ACM International Conference on Multimedia (MM '24), October 28-November 1, 2024, Melbourne, VIC, Australia.* ACM, New York, NY, USA, 9 pages. https://doi.org/10.1145/3664647.3681348

## 1 Introduction

Text-to-Speech (TTS) has experienced significant advancements in recent years, driven by breakthroughs in artificial intelligence, particularly in the field of deep learning. The autoregressive TTS framework, exemplified by the combination of the Tacotron series [25, 31] and WaveNet [17], has significantly improved the naturalness of synthesized speech. Subsequently, non-autoregressive models [10, 12, 21, 23, 24] drastically accelerating the speech synthesis process while retaining high quality. Recently, zero-shot TTS system have emerged [2, 26, 30], pushing the envelope further by synthesizing voices they have never been trained on directly.

Most of these models are capable of generating speech quality close to the human level, but they mainly focus on single-sentence speech modeling. There are many scenarios requiring speech synthesized in a context-aware manner, such as audiobook, conversational speech, and long-form news reading. A straightforward solution is to leverage external information from contextual data into TTS modeling [7, 33, 35]. However, this approach demands a significant amount of high-quality, long-form voice data, which is extremely scarce and hard to obtain for most target speakers. Additionally, given contextual text information, constructing the correlation between it and the current sentence's speech expression is a non-trivial task. As prosody features are commonly used to represent speech expression, many works focus on predicting prosody information from the input text [19, 23, 28] to improve TTS performance. Nonetheless, efforts to model the prosody considering context beyond the current sentence [34] remain limited due to the requirement for long-form data from the target speaker.

In this paper, we propose a Contrastive Context-Speech Pretraining (CCSP) framework to learn cross-modal representations invloving information from both contextual text and current speech expression. As shown in Figure 1, the CCSP model is trained by a large amount of contextual voice data. With no requirement on explicit speaker identification, the CCSP model can leverage as more as possible contextual voice data from different speakers. For

example, Librivox[9] provides huge number of public domain audio-book data read by volunteers from around the world. By leveraging extensive contextual voice data into cross-modal representation learning, the CCSP can align the features of context and speech within a shared space, resulting in enriched feature representations. For instance, the text modality features derived from the surrounding context encapsulate speech representation, and conversely, the speech modality features of the spoken sentence are imbued with contextual information.

After obtaining the pretrained CCSP model, we can inject contextual information into downstream TTS model for any target speaker without considering if there is enough contextual voice data. Figure 1 illustrates the strategic utilization of CCSP model in different stages of TTS model. In the TTS training phase, where paired speech-text data is available, CCSP model is used to generate context-aware speech modality features for the current sentence. These enriched speech modality features enable the TTS system to learn how to produce speech that not only aligns with the content but also reflects the surrounding context. During the inference stage, where the TTS model must generate speech solely from text, the CCSP model comes into play by providing text modality features that capture the speech expression related information. These text modality features effectively replace the speech modality features used in the training phase, resulting in context-aware speech synthesis with a heightened level of expressiveness.

The key contributions of our work can be summarized as:

- We propose a contrastive context-speech pretraining (CCSP) framework to learn cross-modal representations containing both contextual text information and the speech expression of the spoken sentence.
- By integrating cross-modal information from the CCSP model into the downstream TTS system, we can create context-aware voice generation model for any speaker, circumventing the limitation of obtaining sufficient contextual voice data from the target speaker.
- We conduct context-speech retrieval and downstream TTS tasks to evaluate the proposed CCSP model. Experiments demonstrate that the CCSP model is able to learn effective cross-modal representations, and that integrating these learned features enhance the expressiveness of the downstream TTS model, particularly in long-form reading.

## 2 Related Work

### 2.1 Cross-Modal Feature Learning

Cross-modal feature learning is a powerful approach for bridging the semantic gap between various data types, like text and images, or text and audio. By learning representations that capture the underlying relationships across modalities, these models can perform a wide range of tasks that require an understanding of multiple types of input. A representative work of cross-modal feature learning is CLIP [22] (Contrastive Language-Image Pretraining). It leverages a dual-encoder architecture comprising: an image encoder and a text encoder, which are trained to project images and text into a shared embedding space. Using a contrastive loss function [18], CLIP is trained on a diverse and large-scale dataset of image-text pairs to

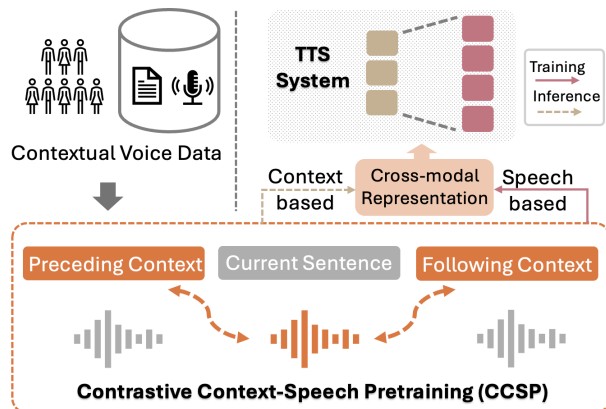

**Figure 1: The overall concept of CCSP and its integration strategy with the TTS system. The CCSP model is trained on abundant contextual voice data without the need for explicit speaker identification. Once the model is pre-trained, the speech-based representation is utilized to guide the training phase of the TTS model. In contrast, the context-based representation is employed during the inference stage.**

maximize the similarity between corresponding image-text pairs while minimizing it for non-corresponding ones.

Similar cross-modal feature learning works between language and speech are wav2vec 2.0 [1], HuBERT [8], and w2v-BERT [4]. These work have underscored the ability to utilize extensive speech data to enhance speech processing tasks, particularly in the domain of ASR (Automatic Speech Recognition). These cross-modal features tend to prioritize the extraction of semantic features from speech while potentially underrepresenting supra-segmental features beyond semantics like prosody. For instance, given the same sentence spoken by different speakers or in different expression manner, the goal of an ASR system is to obtain the same text output. Conversely, as a one-to-many task, TTS systems need to produce correct pronunciation and replicate the spoken way of those words.

CLAP (Contrastive Language-Audio Pretraining) [5, 32] is a cross-modal feature learning framework designed for text and audio, using specialized pre-trained encoders. Compared to the previously mentioned speech encoding approach, the audio encoder [3] utilized in [32] captures a wider range of shifts and patterns by interpreting audio spectrograms as visual images. Additionally, the text input in such language-audio pretraining tasks is typically a description rather than a transcription of the speech, which extends the modeling goal from local semantic understanding to global pattern description to some extent.

### 2.2 Context-aware Text-to-Speech

The development of context-aware TTS systems has been driven by the quest to produce speech that mirrors the nuanced expressiveness of human speech across different contexts. This advancement is crucial for applications that demand high expressiveness, such as audiobook narration, where the system must convey the mood of the story, or in long-form content reading, where maintaining

listener engagement through varied intonation is essential. Traditional TTS systems, while effective in generating intelligible speech, often fall short in capturing the emotional nuances and situational contexts that are intrinsic to natural speech. To bridge this gap, researchers have been exploring various methods to endow synthetic speech with a more dynamic and context-sensitive prosody. This includes techniques in local and global context modeling.

Local context modeling in TTS focuses on information from the textual environment within the sentence or the phonetic details surrounding the speech segment. This includes modeling the nuances of pronunciation, intonation, and timing based on nearby words or segments. For instance, Char2Wav [27] represents an early approach to generate speech from characters by a sequence-to-sequence model that accounts for local phonetic contexts. Recently, Conformer [6] combine convolutional layers for detailed local context modeling and enhances the expressiveness of generated voice for TTS system [16]. CLAPSpeech [36] employs the CLAP framework to learn prosody from text context, which is also focus on modeling local context at the phoneme/word level. These models primarily concentrate on local context modeling without considering broader context information beyond the current sentence.

Global context modeling aims to capture information from an extensive scope beyond the current sentence, potentially encompassing preceding and succeeding contexts or the entire paragraph. This approach enables the TTS system to generate speech that is contextually appropriate and expressive. For instance, cross-sentence information can be utilized by a Bert feature-based contextual encoder [7]. ParaTTS [35] and ContextSpeech [33] consider contextual information from both paragraph text and speech. Additionally, graph-based conversational TTS systems [13, 14] model cross-sentence information by treating them as connected graph nodes. These models involve global contextual information but require parallel contextual voice data from the target speaker.

## 3 Contrastive Context-Speech Pre-training

In this section, we will introduce the Contrastive Context-Speech Pretraining (CCSP) framework, which is illustrated in Figure 2. It comprises three components: the context branch, the speech branch, and the contrastive loss mechanism. The speech branch focuses on processing and understanding the acoustic signals, while the context branch deals with the surrounding textual information. The contrastive loss functions as the bridge between the two, aligning the vector spaces of both modalities to facilitate cross-modal representation learning. The main objective is to learn the associative information between the context and the current speech. In the following subsections, we will provide a detailed exploration of each component within our system.

### 3.1 Context Branch

To avoid concentrating on modeling the transcription-related semantic information during the feature alignment, we use the surrounding context (preceding and following) as the text input and the current sentence's text is omitted. This design is driven by:

- In a TTS system, the input is text and the output is speech. During the TTS modeling process, the current sentence text is provided, carrying with it inherent semantic and prosodic information directly. There exist various explicit modeling

methods designed to predict prosody from the text input. Our focus shift to learn the prosody relationship between surrounding context and the current speech.

- Retaining the current sentence text would cause the cross-modal representation learning to be predominantly influenced by transcription-related semantic information. Our objective is to employ the context as a prompt to generate natural and expressive speech. Therefore, excluding the current sentence text force the modelling to focus on context & speech expression correlation.

As shown in Figure 2, the context branch takes the contexts (preceding $W$ words and following $W$ words) as input and then go through a pretrained text encoder to produce three context embeddings, denoted as $E_P^C$, $E_F^C$, and $E_A^C$. Here, $E^C$ indicates the embeddings produced by context branch. Based on that, $E_P^C$ represents the average text representation derived from the preceding context, $E_F^C$ corresponds to that from the following context, and $E_A^C$ is the combined average text representation obtained from both the preceding and following contexts. The pretrained text encoder we used in our experiments is RoBERTa-base [15].

### 3.2 Speech Branch

*3.2.1 Audio Representation.* In the speech modeling branch, the input speech is initially processed as audio data by the methodology described in [32]. At first, mel-spectrograms are extracted from the variable-length audio samples. These spectrograms are then passed through a feature fusion module to be integrated to a fix length (10s). To be more specific, if the input audio < 10s, we first repeat the input audio and then pad it with zeros. If the input audio > 10s, we downsample the audio to a global input of 10s. Additionally, we extract three separate 10s clips from the beginning, middle, and end $\frac{1}{3}$ of the input to serve as local inputs. These four 10s audio segments are combined into a single feature by the feature fusion method described in [32]. The employed pre-trained audio encoder, HT-SAT model [3], is an audio transformer architecture incorporating a hierarchical structure with a token-semantic module to obtain an effective audio representation, denoted as $E^A$.

*3.2.2 Prosody Representation.* To enrich the cross-modal representation with more supra-segmental information, we incorporate prosody features, pitch and duration, into the speech modeling branch. Initially, we extract frame-level pitch sequence and duration information. The prosody encoder is then utilized to generate positional prosody features as Equation 1. For each frame $i$, $P_i^v$ represents the pitch value. The pitch value is then processed through $PE$, a convolution layer-based pitch encoding module, transforming the scalar pitch value into an $m$-dimensional vector. Additionally, $P_i^d$ indicates the phoneme index corresponding to frame $i$, and $PS(P_i^d)$ retrieves the positional embedding according to the method established by [29]. By summing these components, we acquire the positional prosody feature, $E_i^P$.

$$E_i^P = PE(P_i^v) + PS(P_i^d) \tag{1}$$

$$E_\mathbf{x}^P = GRU(E_{x_1}^P, E_{x_2}^P, ..., E_{x_T}^P) \tag{2}$$

$$E_\mathbf{x}^S = proj([E_\mathbf{x}^P | E^A]) \tag{3}$$

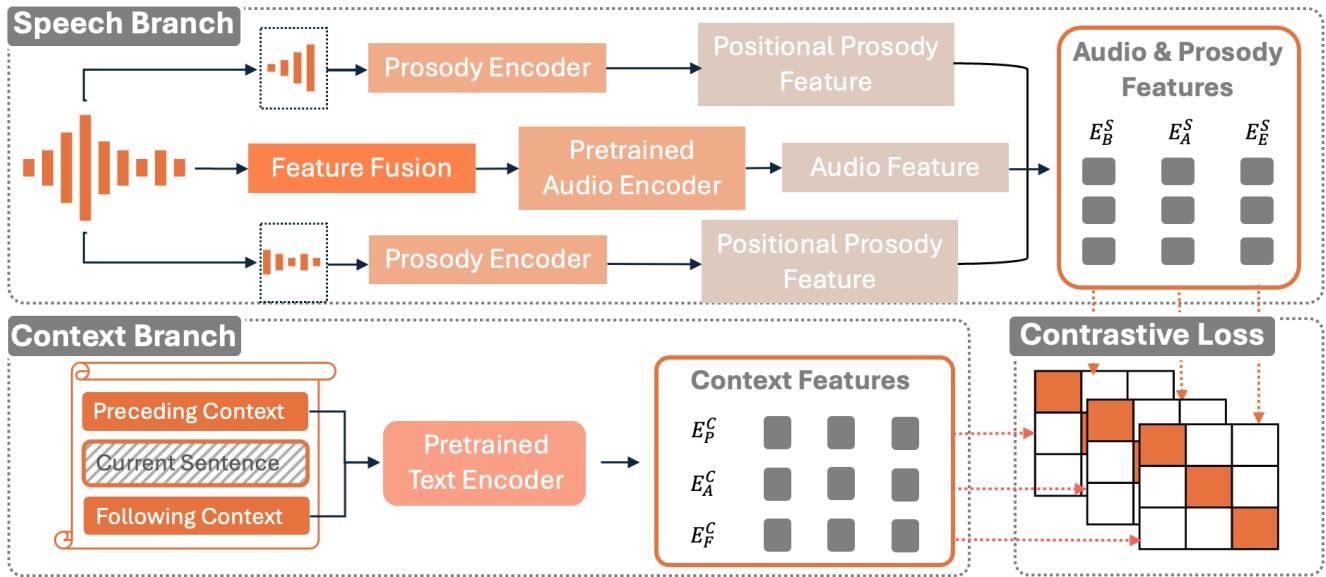

Figure 2: The Overall Framework of CCSP. The Speech Branch takes audio as input, goes through a prosody encoder and a pre-trained audio encoder, and then the outputs are concatenated to form the Audio & Prosody Features. The Context Branch takes the preceding and following context as input, goes through a pre-trained text encoder to obtain the Context Features. Finally, a contrastive loss function will be used to align the features from different modalities.

After obtaining the frame-level positional prosody features, we employ a GRU layer to derive a global prosody feature, denoted as $E_{\mathbf{x}}^P$, for speech segment $\mathbf{x}$ in Equation 2. $\{x_1, x_2, ..., x_T\}$ are frames belonging to $\mathbf{x}$. Subsequently, the Audio & Prosody Feature for speech segment $\mathbf{x}$, $E_{\mathbf{x}}^S$, is acquired by concatenating the prosody feature ($E_{\mathbf{x}}^P$) with the Audio feature ($E^A$), followed by a projection as in Equation 3. As depicted in Figure 2, three Audio & Prosody Features are generated: $E_B^S$, $E_A^S$ and $E_E^S$, corresponding to speech segment "Beginning", "All" and "End". An assumption here is that the preceding context predominantly influences the beginning part of the current sentence, whereas the following context mainly impacts the end part of the current sentence. To be more specific, speech segments $B$, $A$ and $E$ refer to the first $s$ seconds, the entire sentence, and the last $s$ seconds, respectively.

### 3.3 Contrastive Learning

To effectively learn representations that integrate information from dual modalities, it is essential to align the feature spaces of the respective modal representations. Building on the approaches of [22, 32], we simultaneously train the speech and context branches with the objective of maximizing the cosine similarity between the modality representations. Within a batch size of $N$, we generate $N^2$ context-speech pairings. Each pair is processed through its corresponding branch, resulting in three textual features ($E_P^C$, $E_A^C$, and $E_F^C$) and three speech features ($E_B^S$, $E_A^S$, and $E_E^S$). This process forms $N^2$ pairings each for ($E_P^C$, $E_B^S$), ($E_A^C$, $E_A^S$), and ($E_F^C$, $E_E^S$). The goal for these pairings is to optimize the similarity scores for the $N$ intra-sample pairs while reducing similarity for the $N(N-1)$ inter-sample pairs. The loss function, $L$, based on symmetric cross

entropy ($f_{SCE}$), is detailed in Equation 4, encapsulating this alignment objective.

$$
\begin{aligned}
L = & f_{SCE}([E_{A,1}^S, ..., E_{A,N}^S], [E_{A,1}^C, ..., E_{A,N}^C]) \\
& + f_{SCE}([E_{B,1}^S, ..., E_{B,N}^S], [E_{P,1}^C, ..., E_{P,N}^C]) \\
& + f_{SCE}([E_{E,1}^S, ..., E_{E,N}^S], [E_{F,1}^C, ..., E_{F,N}^C])
\end{aligned}
\tag{4}
$$

$$
\begin{aligned}
& f_{SCE}([E_{\mathbf{x},1}^S, ..., E_{\mathbf{x},N}^S], [E_{\mathbf{x},1}^C, ..., E_{\mathbf{x},N}^C]) \\
& = \frac{1}{2N} \sum_{i=1}^{N} \Big( log \frac{exp(E_{\mathbf{x},i}^S \cdot E_{\mathbf{x},i}^C / \tau)}{\sum_{j=1}^{N} exp(E_{\mathbf{x},i}^S \cdot E_{\mathbf{x},j}^C / \tau)} \\
& + log \frac{exp(E_{\mathbf{x},i}^C \cdot E_{\mathbf{x},i}^S / \tau))}{\sum_{j=1}^{N} exp(exp(E_{\mathbf{x},i}^C \cdot E_{\mathbf{x},j}^S / \tau))} \Big)
\end{aligned}
\tag{5}
$$

## 4 Downstream Task

### 4.1 Context-Speech Retrieval

As the CCSP model is designed to learn cross-modal representations between context and speech, cross-modal retrieval task is a straightforward way to verify the effectiveness of the learned representation. **Context-to-Speech Retrieval** task is selecting the most relevant context instance from a set $\{E_{\mathbf{x},1}^C, E_{\mathbf{x},2}^C, ...\}$ for a given speech representation, $E_{\mathbf{x},i}^S$. Conversely, **Speech-to-Context Retrieval** task is finding the most corresponding speech statement from a collection $\{E_{\mathbf{x},1}^S, E_{\mathbf{x},2}^S, ...\}$, given a context instance $E_{\mathbf{x},i}^C$. Compared with evaluation on TTS system, these Context-Speech Retrieval tasks leverage the cross-modal representations produced by the

CCSP model directly, obviating the need for an additional modeling process. Therefore, these two retrieval tasks can serve as a rapid verification method to help us validate and adjust the parameters of the CCSP model. They also offer an objective benchmark for further development and refinement.

## 4.2 Expressive Text-to-Speech

Given the pre-trained CCSP model, we utilize the NaturalSpeech 2 (NS2)[26] framework as the backbone TTS system to evaluate whether the pre-trained model enhances the expressiveness of the TTS model by delivering effective contextual information. Figure 3 illustrates the NS2 model's architecture and the way to inject cross-modal representations from CCSP model. The components of a baseline NS2 model can be described as: 1) Phoneme Encoder. This component encodes the input text into a phoneme sequence, which serves as a linguistic prior for the model. 2) Prosody (pitch/duration) Predictors. Alongside the phoneme encoder, prosody predictors estimate the duration of each phoneme and the pitch contour for the speech synthesis process. These components are crucial for achieving natural prosody and intonation in the synthesized speech. 3) Audio Codec. The audio codec is composed by a encoder, a residual vector-quantizer (RVQ) and a decoder. Codec encoder downsample the input audio and the RVQ converts the reuslt to a sequence of latent vectors. While the codec decoder reconstructs the speech waveform from these latent vectors. 4) Diffusion Model. The diffusion model generates the sequence of latent vectors conditioned on the phoneme sequence and the outputs from the duration and pitch predictors. 5) Speech Prompt Encoder. To facilitate the zero-shot capabilities, the model employs in-context learning, which allows the model to adapt to new speaker characteristics by being conditioned on a speech prompt. The speech prompt is randomly segmented from the ground truth speech during training. For inference, we utilize a speech segment less than 10 seconds from an unseen speaker as reference to generate synthesised samples for testing.

The speaker embedding generated by the speech prompt encoder will influence both prosody feature prediction and sound reconstruction. Since prosody information is an essential part of the expressiveness of the generated voice, we incorporate cross-modal representations from CCSP into the speaker embedding to jointly affect the prosody feature predictor. More specifically, $E_A^S$ from the Speech Branch of CCSP serves as the injected embedding during training, while $E_A^C$ from the Context Branch is used during inference. This approach is taken because ground-truth speech is always available during TTS training, whereas context information is provided or can be generated by a GPT-service [20] during inference. Based on this injection strategy, we can conduct effective contextual modeling for any target speaker even without contextual voice data.

## 5 Evaluation

### 5.1 Dataset

**LibriVox** LibriVox [9] is a unique online project that harnesses the power of volunteers to create free, public domain audiobooks from texts that are also in the public domain. As audiobook data, Librivox contains a wealth of contextual voice data, which aligns with our requirements for training the CCSP model. We downloaded and processed approximately 15,000 hours of data, which

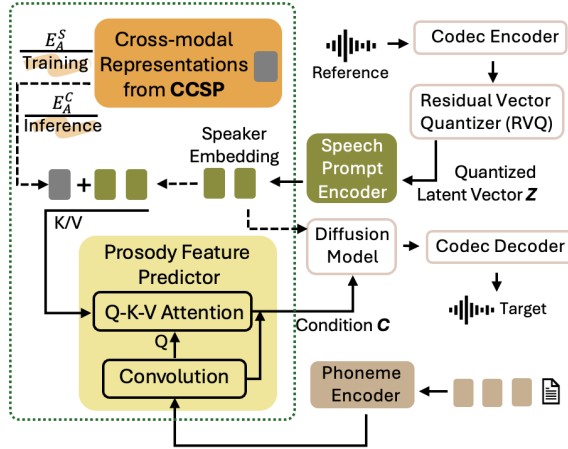

**Figure 3: NS2 Model with CCSP Features. The CCSP embeddings are added to the speaker embedding to exert a combined effect on the prosody feature predictor. The $E_A^S$ from the Speech Branch of CCSP is injected during training, while the $E_A^C$ from the Context Branch is used during inference.**

included data segmentation, alignment, and feature extraction. In our experiments, we utilize two datasets: the complete ~15,000-hour collection, **LibriVox-B**, and a subset consisting of ~1,500 hours, **LibriVox-S**, primarily to compare the impact of dataset size on the performance of the CSSP model.

**LibriTTS-R** We use LibriTTS-R [11] as the target data for our TTS task as it is a multi-speaker TTS corpus without ground-truth contextual information. LibriTTS-R is an enhanced version of the original LibriTTS [37] corpus. LibriTTS is a text-to-speech corpus crafted from LibriVox audiobooks. It is optimized for TTS research with features like 24kHz audio for better spectral content, precise text-to-audio alignment, and a diverse range of speakers and accents, primarily in English. LibriTTS-R improves the audio quality by applying speech restoration techniques to the original LibriTTS recordings, which contains 585 hours of data from 2,456 speakers.

### 5.2 Evaluation Metrics

**Objective Test** We evaluate the performance of both Context-to-Speech Retrieval and Speech-to-Context Retrieval tasks using the objective metric *mAP@10* (mean Average Precision at rank 10). The computation of mAP@10 is presented in Equation 6, where $q$ is the index of a query sample, $Q$ is the total number of test samples, and $p_q$ indicates the rank position of the correctly matched sample for the $q^{th}$ query after sorting by their corresponding logit values. For our experiments, we employed a test set reserved from the LibriVox-B dataset, comprising $Q = 1024$ samples.

$$mAP@10 = \frac{1}{Q} \sum_{q=1}^{Q} \frac{1}{p_q + 1} * rel(q) \tag{6}$$

$$rel(q) = \begin{cases} 1, \text{ if } p_q < 10 \\ 0, \text{ else} \end{cases} \tag{7}$$

To assess an expressive TTS system, we perform an objective test by computing the *Normalized Pitch Dynamic Score*. This metric captures the mean pitch variation at the syllable level and normalizes it to a 0-1 range using a sigmoid function. A greater normalized pitch dynamic score indicates a richer variation in pitch sequence, suggesting a higher level of expressiveness in the synthesized speech.

**Subjective Test** For a perceptual assessment of the impact brought by CCSP features to the TTS model, we conduct subjective tests, including the *Preference Test* and *Comparative Mean Opinion Score (CMOS)*. The Preference test is a blind listening test that assesses whether listeners can detect a difference between two audio samples and express a preference for one over the other. The outcome of this test is expressed in percentages, showing the proportion of test sample pairs where listeners preferred either system A or system B, along with the percentage of cases where there was 'No preference'. For the CMOS test, listeners compare the expressiveness of speech generated by two TTS systems. Unlike the preference test, CMOS test requires listeners to not only choose which sample they consider better but also to rate the extent of its superiority or preference on a scale from -3 to +3, with the reference samples assigned a score of 0. In our experimental setup, we engaged 9 native speakers to do the evaluation in a quiet environment.

We create two test sets, a *sentence-based test set* with 50 cases and a *paragraph-based test set* with 30 cases. Each paragraph case is composed by 2 to 5 sentences. As paragraph tests involve longer test samples, which add to the complexity for reviewers, we offer higher rewards to participants of paragraph-based evaluations. The scripts for these tests were generated by the GPT-service [20], ensuring a unbiased selection of content for evaluation.

## 5.3 Analysis on CCSP Model

**Context Length** We conducted experiments to study the impact of different context lengths on the performance of the CCSP model.

*Context Length during Training.* Intuitively, a longer context input would provide the model with more information, potentially leading to improved results. However, the observations from Figure 4 challenge this assumption. The figure depicts the mAP@10 metric on the LibriVox-B dataset, with the left graph and right graph representing the Context-to-Speech retrieval task and the Speech-to-Context retrieval task respectively. The context length of 20 refers to a maximum limitation of 20 preceding words and 20 following words during the training stage. From the result, we found that both retrieval tasks favored a 20-context instead of the longer one, 80-context, along with the increase of training epochs. There could be several reasons for these observations. 1) While providing more context has the potential to furnish additional information for the model, the impact of this text tends to decrease as the distance from the target sentence grows, especially when we attempt to model the relationship between the context and the speech expression. 2) The actual context length of each sentence in the dataset might be limited. In Figure 6, we present the distribution of the actual context length on the LibriVox-B dataset. The distribution reveals that a significant majority of sentences (> 40%) possess a context length less than 20. This observation may offer a plausible explanation for the experimental results.

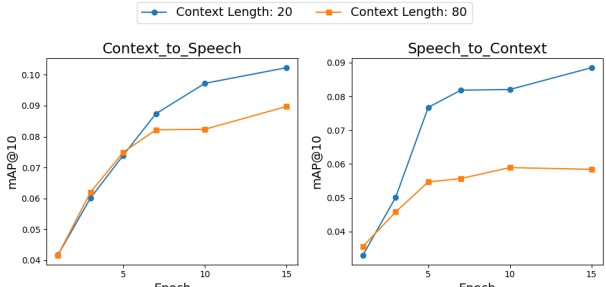

**Figure 4: Analysis on Context Length - Training. Both retrieval tasks favored the 20-context instead of the 80-context along with the increase of training epochs.**

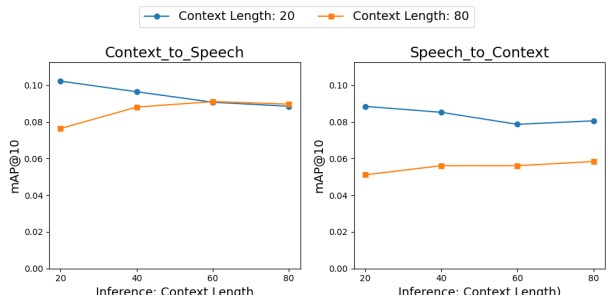

**Figure 5: Analysis on Context Length - Inference. 1) Matched training and inference context lengths leads to optimal model performance. 2) A 20-20 (training-inference) context length configuration excels over other pairings.**

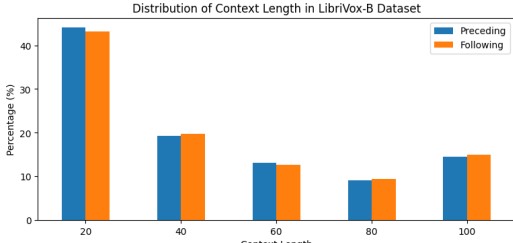

**Figure 6: Context Length Distribution of LibriVox-B Dataset. Over 40% of sentences have a context length shorter than 20.**

*Context Length during Inference.* Additionally, we investigated the impact of context length employed during the inference stage. Figure 5 demonstrates the results for two models, trained with context lengths of 20 and 80, respectively. It is apparent that both models perform optimally when the context length during inference matches that used at the training phase. Overall, The combination of a 20-20 (training-inference) context length outperformed other configurations. Considering these findings, we decided to proceed with the 20-20 context length configuration in our subsequent experiments. This is based not only on the performance but also on the advantage of saving memory.

**Prosody Length**. To enclasp more speech expression related information, we incorporate prosody features in speech branch to enhance the cross-modal representation learning. As introduced

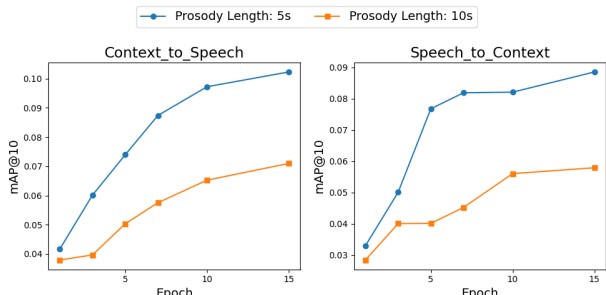

**Figure 7: Analysis on Prosody Length. 5-second configuration significantly outperforms the 10-second setting.**

in Section 3.2.1, the speech data with variant length are processed in a fixed duration of 10 seconds through a feature fusion function. The optimal duration of prosody features remains an additional parameter for analysis. As detailed in Section 3.2.2, three types of prosody features are extracted from the current sentence: "Beginning", "All", and "End". These are then fused with the audio feature. The "Beginning" prosody feature aligns with the preceding context and the "End" prosody feature aligns with the following context. Figure 7 contrasts the performance of CCSP models utilizing different prosody feature lengths, specifically 5s and 10s. The 10-second duration matches the length of the processed audio feature, whereas the 5-second duration is half as long. As shown in Figure 7, the 5-second configuration significantly outperforms the 10-second setting in both Context-to-Speech and Speech-to-Context retrieval tasks. This suggests that: 1) longer prosody representations might present increased complexity and difficulty for effective modeling, 2) the influence of context on prosody is primarily concentrated around the immediate context—meaning the beginning part of the speech is most influenced by the preceding context, while the end part of the speech is most affected by the following context. Based on these findings, we will adopt a 5-second length for prosody feature in our subsequent experiments.

## 5.4 Analysis on Expressive TTS system

**Experimental Setting**. As we mentioned in Section 4.2, we use NS2 as our backbone TTS model, which is a state-of-the-art zero-shot TTS system. The configuration of the NS2 model is consistent to that presented in the original publication. The additional contextual embeddings are processed through a 512x512 linear projection layer before being added to speaker embedding. We employ two baseline NS2 model with different training strategies:

- *NS2-A* model, trained from scratch by the LibriTTS-R dataset.
- *NS2-B* model, initially trained on the Librivox-B dataset and subsequently fine-tuned on the LibriTTS-R dataset. It serves as an enhanced baseline, employing an equal amount of data as the pre-trained CCSP model, to validate the proposed framework's effectiveness while isolating the variable of additional data input.

The NS2+CCSP model, the NS2 model with injection of CCSP features, is trained on the LibriTTS-R dataset in the same manner as the NS2-A model. Within this framework, the CCSP model serves

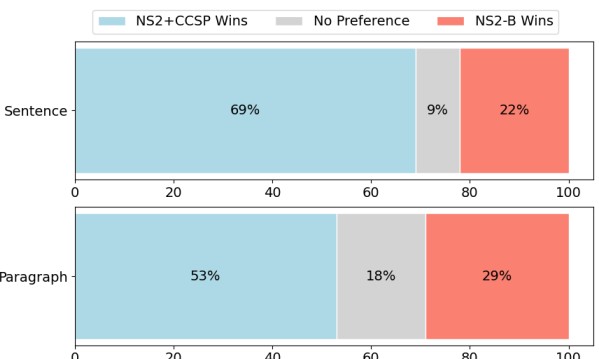

**Figure 8: Preference Test on Sentence / Paragraph Test Set. NS2+CCSP model is favored over the baseline model in both.**

**Table 1: Normalized Pitch Dynamic Score. NS2-B model slightly surpasses NS2-A; NS2+CCSP notably excels, highlighting influence from CCSP.**

| TEST SET | NS2-A | NS2-B | NS2+CCSP |
|---|---|---|---|
| SENTENCE | 0.27 | 0.28 | 0.36 |
| PARAGRAPH | 0.28 | 0.31 | 0.39 |

**Table 2: CMOS on Sentence / Paragraph Test Set. The NS2+CCSP model attains CMOS improvements in both.**

| TEST SET | NS2-B | NS2+CCSP |
|---|---|---|
| SENTENCE | 0.000 | 0.493 |
| PARAGRAPH | 0.000 | 0.146 |

as a feature generator. It is trained on the LibriVox-B dataset and produces features with a dimensionality of 512.

**Experimental Analysis**. We conduct both objective test and subjective test to do evaluation on the NS2+CCSP model.

*Objective Test*. Objective evaluations were conducted on three models, NS2-A, NS2-B and NS2+CCSP,by calculating the *Normalized Pitch Dynamic Score*. Results presented in Table 1 cover both sentence-based and paragraph-based test sets. The NS2-B model shows a slight improvement in score over the NS2-A model, which indicates the prosody variance improved by data augmentation. Notably, the NS2+CCSP model demonstrates a substantial increase in score. This suggests that while data augmentation through adaptation provides a modest enhancement, the incorporation of the CCSP model, which leverages cross-modal contextual information from extensive datasets, significantly enriches the expressiveness of the text-to-speech (TTS) synthesis.

*Subjective Test*. Building on the objective analysis, subjective evaluations were conducted to further compare the performance of the NS2-B and NS2+CCSP models. These evaluations included a preference test and a Comparison Mean Opinion Score (CMOS) test. Figure 8 presents the preference scores for the subjective evaluation of the NS2+CCSP and NS2-B models. In the sentence-based test set, the NS2+CCSP model was preferred 69% of the time, with a 'No Preference' response at 9%, and the NS2-B model preferred

**Table 3: Ablation Study by CMOS Test. Our ablation studies are conducted from three aspects: model framework, input data construction and data size.**

| | |
|---|---|
| NS2+CCSP | 0.000 |
| NS2+CCSP_WITHOUT-PROSODY | -0.094 |
| NS2+CCSP_CURTEXT | -0.084 |
| NS2+CCSP_LIBRIVOX-S | -0.204 |

22% of the time. For the paragraph-based test set, the preference scores are 53%, 18%, 29% correspondingly. Both of them suggest the NS2+CCSP model is favored over the baseline model in the listener preference tests. Table 2 presents the CMOS results, indicating that the NS2+CCSP model attains CMOS improvements of 0.493 and 0.146 for the sentence-based and paragraph-based test sets, respectively. It is noteworthy that the enhancement is more pronounced in the sentence-based test set than in the paragraph-based set, which can be attributed to the intrinsic challenges of comparing and evaluating the more complex, longer content. These results affirm the proposed model's overall advancements in producing more natural and expressive synthesized speech from sentences to paragraphs.

## 5.5 Ablation study

In this section, we delve into the efficacy of the individual components of the CCSP model through a series of ablation study. Table 3 presents the CMOS test outcomes comparing variants of the NS2+CCSP model to the proposed model. Notice that the test set used here is the sentence-based one.

**Speech Branch**. Figure 9 shows the performance of CCSP model on the Context-to-Speech and Speech-to-Context retrieval tasks, focusing on two variants: one without the audio feature and one without the prosody feature. 1) *Without audio feature*. The variant with only the prosody feature in the speech branch shows a marked decrease in performance. It highlights the significant role of the pre-trained audio encoder, which benefits from large-scale data training. 2) *Without prosody feature*. The variant without prosody feature retains the original CLAP structure performs comparable to the standard CCSP model during the initial stages of training. But after five epochs, the CCSP model that integrates the prosody feature demonstrates its superiority. The CMOS regression (-0.094) between NS2+CCSP and NS2+CCSP_without-prosody models further verifies the effectiveness of integrating of prosody features.

**Context Branch**. We examine the effectiveness of using surrounding context as the text modality input of cross-modal representation learning framework by reverting the context back to the current speech's transcription. The CMOS test results (-0.084) for the variant NS2+CCSP_curtext model in Table 3 confirms the significance of broader contextual information for enhancing TTS expressiveness.

**Data Size**. Figure 10 illustrates the advantages of scaling up data size for retrieval tasks. Subsequently, we developed a corresponding variant, the NS2+CCSP_LibriVox-S model to compare with the NS2+CCSP model. This variant use cross-modal features from CCSP model trained by a smaller dataset, LibriVox-S, which is described in Section 5.1. The CMOS test result (-0.204) in Table 3 shows that the cross-modal contextual embeddings, derived from the CCSP model

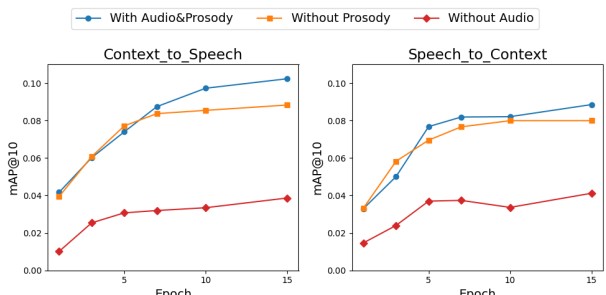

**Figure 9: Ablation Study on Speech Branch. Both audio encoding and prosody encoding in the speech branch improves the CCSP's effectiveness on retrieval tasks.**

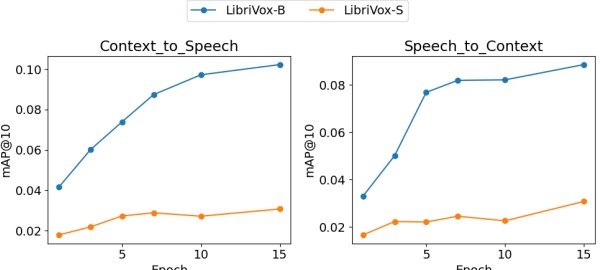

**Figure 10: Analysis on Data Size. Increasing data size notably enhances performance on retrieval tasks.**

trained on a larger dataset, significantly enhance the expressiveness of the TTS system.

## 6 Conclusion

In this paper, we present the Contrastive Context-Speech Pretraining (CCSP) model, designed to enhance expressive TTS systems. Our pretraining framework focuses on learning effective cross-modal representations that capture the relationship between global contextual information and the current speech expression. Through the utilization of abundant contextual speech data and explicit prosody modeling, the generated features demonstrate their efficacy in both the Context-to-Speech and Speech-to-Context retrieval tasks. By strategically integrating these cross-modal representations into a downstream TTS system, we observe a significant improvement in the expressiveness of the generated speech, as evidenced by both subjective test and objective test. Overall, our proposed CCSP framework proves beneficial for expressive TTS systems, particularly in scenarios where there is a dearth of sufficient high-quality, long-form speech data for the target speaker.

*Future Work and Impact Statements.* • While the CCSP framework has shown promising results in learning cross-modal representations and enhancing the expressiveness of synthesized speech, this work can be further refined by 1) improved fine-grained control over expressions and 2) broadening its application to various downstream tasks such as speech-to-speech generation and speech style annotation tasks. • This work enhances TTS expressiveness with limited target speaker data, which promises improvements in user experience across digital media and accessibility. However, it also raises ethical concerns like creating challenges in authenticity verification and potentially facilitating misinformation.

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
