# OpenReview forum: "Contrastive Context-Speech Pretraining for Expressive Text-to-Speech Synthesis"
_acmmm.org/ACMMM/2024/Conference — MM2024 Poster_

### Official Review · Reviewer_hDbG · 2024-05-16

**Rating:** 4
**Confidence:** 3

**Summary:**

This paper introduces a context-aware voice pretraining model leveraging contrastive learning for expressive Text-to-Speech (TTS) synthesis. The model leverages contextual voice data without explicit speaker labels to capture the intricate relationship between context and speech expression. This enables the generation of expressive speech by integrating cross-modal features of context and speech into the TTS model, particularly beneficial when target speaker data is scarce.

**Strengths:**

In both validation tasks where the proposed pretraining model is 1) applied to Context-Speech retrieval tasks for parameter optimization and 2) integrated with a zero-shot TTS system to assess the assistance of learned cross-modal features in expressive TTS modeling, improvements over baseline models were obtained. Both objective and subjective evaluations were carried out in the assessment of the TTS expressiveness. In addition, the paper is well organized and is clearly written.

**Limitations:**

The paper uses NS2 as a baseline system and mentions that NS2 is a state-of-the-art zero-shot TTS system. However, there is no reference to NS2 in the paper. Also, a comparison with other TTS system which can synthesize expressive speech could be carried out. The current comparison with NS2 is limited.

**Suitability:**

2

---

### Official Review · Reviewer_muvW · 2024-05-24

**Rating:** 2
**Confidence:** 4

**Summary:**

This work leverages contrastive learning to extract corresponding prosodic information from contextual text, aiming to improve the performance of long-text TTS models.

**Strengths:**

1.Although contrastive learning has been employed for prosody prediction in several works, such as ClapSpeech[1], there has been no prior attempt to predict prosody from long contextual text. Thus, the method itself is not highly innovative, but it is applied in a new tts scenario.


2.The authors provide a detailed analysis of the CCSP model in the ablation study section.


[1] Ye Z, Huang R, Ren Y, et al. Clapspeech: Learning prosody from text context with contrastive language-audio pre-training[J]. arXiv preprint arXiv:2305.10763, 2023.

**Limitations:**

1. I am concerned about the actual utility of the CCSP module. Methodologically, does the audio side of the CCSP module—specifically
${E_{B}}^{S}$ , ${E_{A}}^{S}$ and ${E_{E}}^{S}$ contain semantic information? If so, I am unsure if contrastive learning will focus on prosody alignment rather than semantic alignment. Experimentally, the ablation study from lines 840 to 853 suggests that the absence of prosodic representation has a minor impact, while the absence of pretrained audio encoder representations has a significant effect. Additionally, the MAP values in the retrieval task are quite low, casting doubt on the efficacy of the CCSP module.


2. Upon listening to the author's demo, I noticed no significant differences, and it appears that no ground truth (GT) was provided. Furthermore, the demo includes only two sets of audio samples. I recommend the authors provide more audio samples (e.g., 30 sets) during the rebuttal period for a more thorough comparison.


3. Evaluating the CCSP module, a plug-and-play component, solely within the NaturalSpeech2 model is insufficient. I am curious why the authors chose a zero-shot TTS baseline rather than adding the CCSP module to a standard TTS model. I suggest the authors include more baselines, such as VALLE, FastSpeech2, and VITS, to demonstrate the effectiveness of the CCSP module.

4. The writing quality needs improvement. Some simple concepts are explained in a convoluted manner. For instance, the CCSP module, which is essentially a straightforward contrastive learning approach for contextual text, is overcomplicated. Additionally, in line 453, the model is referred to as NS2, which might confuse reviewers unfamiliar with zero-shot TTS. Similarly, in line 327, the PE module is not explained, even though it can be inferred as an encoder.


5. In the retrieval domain, it is common to calculate both recall and MAP. I recommend the authors include recall-related experiments for further validation.


6. For fairer comparisons, I suggest the authors conduct tests on public datasets.

7. I am curious whether the authors plan to open-source their code.


8. In Table 3, when the CCSP module is added to TTS, I may have misunderstood the specific content of
${E_{B}}^{S}$ and ${E_{B}}^{S}$ . Could the authors provide examples about ${E_{B}}^{S}$ and ${E_{B}}^{S}$ during the rebuttal period?


9. The introduction mentions that the CCSP module is designed to guide prosody in long texts. However, the Sentence/Paragraph Test Set experiment (Table 2) shows that the effect is weaker for paragraphs than for sentences. This may be due to unstable prosody in paragraphs, potentially indicating that the CCSP module's effectiveness in guiding prosody for long texts is limited.

**Suitability:**

3

---

### Official Review · Reviewer_wkCt · 2024-05-24

**Rating:** 5
**Confidence:** 3

**Summary:**

This paper proposes a novel context-aware speech pre-trained model for expressive TTS based on contrastive learning. It is designed to learn cross-modal representations between context and speech and leverages these representations to enhance the expressiveness of the TTS model. When applying it to NaturalSpeech2 framework, it improves the expressiveness of speech even with limited data. The experimental results show that the proposed model significantly enhances the expressiveness of synthesized speech.

**Strengths:**

Novelty: This paper proposes a simple but effective method to enhance current SOTA  zero-shot TTS system. It is novel that it combines the cross-modal retrieval task and TTS.
Theoretical approach: The proposed model is designed to learn effective cross-modal representations that capture the relationship between global contextual information and the current speech expression and makes a significant improvement in the expressiveness of the generated speech. It's of a certain level of technical depth.
Clarity: The article is clearly stated and the sections are closely connected logically
Adequate evaluation: It conducts sufficient objective and subjective tests to evaluate the performance of the proposed model in downstream. Also conduct detailed analysis on context-length and prosody-length which which are directly related to the proposed model.
Application: It shows a potention to breaking the bottleneck of data amount in Zero-shot TTS. And it can be applied to other downstream work related to long text or speech in future.

**Limitations:**

1. NS2 is an abbreviation. When it first appeared, the full name was not described and no citation was added.
2. The font size in Figure 2 is too large and inconsistent with the article.
3. About the selection of the base models. In order to prove that this pre-trained model is  flexibility in the TTS area, the selection of backbone TTS models should be more diverse. Then the corresponding baseline models will be various and the improvement will be more evident.

**Suitability:**

3

---

### Official Review · Reviewer_hPvR · 2024-05-26

**Rating:** 5
**Confidence:** 2

**Summary:**

This paper introduces a CCSP framework, which adopts a contrastive learning approach to learn cross-modal information between contextual representation and speech representation, thereby generating higher-quality expressive speech. It is particularly valuable in situations characterized by limited target speaker data and the need for lengthy expressive readings. Experimental results also demonstrate the effectiveness of this pre-training framework across two downstream tasks: Context-Speech retrieval and zero-shot TTS system.

**Strengths:**

- The paper is well-structured and the motivation is solid.
- The reference and surveys are adequate.
- The authors conduct extensive experiments with ablations and analysis to show the strong foundation.

**Limitations:**

For context-aware text-to-speech, synthesizing long texts presents a significant challenge. Therefore, I'm particularly interested in the long-form expressive TTS capabilities of the CCSP model. I do notice that this paper covers paragraph tests composed of 2-5 sentences. However, it would be more convincing and supportive if it could be longer in most cases.

**Suitability:**

3

---

### Meta-Review · Area_Chair_Hb7J · 2024-07-02

**Recommendation:** Accept (Poster)
**Confidence:** 2

**Metareview:**

This paper introduces a contrastive learning approach for expressive speech synthesis representation learning. The goal is to learn cross-modal information between contextual representation and speech representation, thereby generating higher-quality expressive speech. It is particularly valuable in situations characterized by limited target speaker data and the need for lengthy expressive readings. Experimental results also demonstrate the effectiveness of this pre-training framework across two downstream tasks: Context-Speech retrieval and zero-shot TTS system. Reviewers are somewhat split over the paper, but attribute some novelty (contrastive learning for representation learning in a synthesis context). Some open questions remain even after rebuttal, but authors are working to add further answers (e.g. evaluation results on longer utterances) to the final manuscript, which would address most concerns. Still, since results are not yet available, I cannot give a strong recommendation.

Reasons to accept:
- This paper proposes a simple but effective method to enhance current SOTA zero-shot TTS systems.
- The paper is well-structured and the motivation is solid. The references and discussion of related work is appropriate, the paper is over all well written.
- The authors conduct extensive experiments with ablations and analysis to show the contributions.

Reasons to reject:
- Reviewer muvW especially (also hPvR) raises several questions that have no been addressed in the rebuttal. Authors indicate they will have more results to share in the camera ready copy (and improve the writing, share code), but as of now we need to treat these questions as unresolved.